# Planned Physical Workload in Young Tennis Players Induces Changes in Iron Indicator Levels but Does Not Cause Overreaching

**DOI:** 10.3390/ijerph19063486

**Published:** 2022-03-15

**Authors:** Piotr Żurek, Patrycja Lipińska, Jędrzej Antosiewicz, Aleksandra Durzynska, Jacek Zieliński, Krzysztof Kusy, Ewa Ziemann

**Affiliations:** 1Faculty of Physical Culture Gorzow, Poznan University of Physical Education, 66-400 Poznan, Poland; p.zurek@awf-gorzow.edu.pl; 2Institute of Physical Education, Kazimierz Wielki University in Bydgoszcz, 85-064 Bydgoszcz, Poland; patlipka@gmail.com; 3Department of Bioenergetics and Physiology of Exercise, Medical University of Gdansk, 80-211 Gdańsk, Poland; jedrzej.antosiewicz@gumed.edu.pl; 4Faculty of Health Sciences, Lomza State University of Applied Science, 18-400 Łomża, Poland; adurzynska@pwsip.edu.pl; 5Department of Athletics, Strength and Conditioning, Poznan University of Physical Education, 61-871 Poznan, Poland; jzielinski@awf.poznan.pl (J.Z.); kusy@awf.poznan.pl (K.K.)

**Keywords:** sport performance analysis, exercise testing, hepcidin, heat shock protein, overtraining, fatigue, adolescent, inflammation

## Abstract

The current study aimed to examine the impact of the training load of two different training camps on the immunological response in tennis players, including their iron metabolism. Highly ranked Polish tennis players, between the ages of 12 and 14 years, participated in two training camps that were aimed at physical conditioning and at improving technical skills. At baseline and after each camp, blood samples were analyzed, and the fatigue was assessed. The levels of pro- and anti-inflammatory indicators, iron, and hepcidin were determined. The levels of the heat shock proteins, (Hsp) 27 and 70, were also measured. All the effects were evaluated using magnitude-based inference. Although the training camps had different objectives, the physiological responses of the participants were similar. The applied programs induced a significant drop in the iron and hepcidin levels (a small-to-very-large effect) and enhanced the anti-inflammatory response. The tumor necrosis factor α levels were elevated at the beginning of each camp but were decreased towards the end, despite the training intensity being medium/high. The changes were more pronounced in the female players compared to the male players. Altogether, the results suggest that low-grade inflammation in young tennis athletes may be attenuated in response to adequately designed training. To this end, the applied physical workload with a controlled diet and rest-controlled serum iron levels could be the marker of well-designed training.

## 1. Introduction

Long-lasting exercise leads to adaptive changes and improved physical performance in professional athletes, regardless of age. However, in the case of long-lasting physical workloads (chronic training), the effect can be the opposite [1]. Therefore, to prevent hormonal and immunological disruptions, the training workload has to be adjusted accordingly to balance the pro- and anti-inflammatory responses. Otherwise, such disruptions can trigger a spectrum of underperformance conditions, including functional overreaching, or even overtraining [2,3]. The diagnosis of fatigue in tennis is much more difficult than in other sports, mainly because tennis players tend to train individually and the circumstances during tennis matches are largely unpredictable (for example, the number of games and sets played or changing weather conditions). Mood deterioration is one of the symptoms of a disturbed stress–regeneration balance (in overreaching or overtraining), which, together with physical performance, is particularly significant in tennis [4].

The number of studies on the subject, especially concerning young tennis players and blood assessments, is limited. Witek and coworkers [5] evaluated changes in the myokine, the heat shock protein (Hsp), and the growth factor levels in highly ranked adult male tennis players in response to the physical workload during the competitive season, and their correlations with the match scores. The authors reveal a significant increase in the interleukin (IL) 6 levels, which was inversely correlated with the number of lost games. Moreover, elevated concentrations of alpha tumor necrosis factor (TNFα) were registered after the tournament season among high professional senior tennis players [6]. The effect of the physical workload on the synthesis of Hsp27 and Hsp70 that was imposed by a controlled and planned training camp was verified in another study [7]. The authors found that, after the tournament season, the tennis players experienced overreaching syndrome, which is characterized by low Hsp27 levels and high Hsp70 levels, and by elevated levels of the proinflammatory cytokines, IL-1β and TNF-α. HSPs are a family of proteins that regulate protein homeostasis, maintain normal cellular function, and are expressed under different stress conditions, including also exercise. These proteins protect against the aggregation of aberrantly folded proteins and promote their return to their native confirmations. [8]. The diminished Hsp27 levels that were determined at the end of the tournament season indicate that the physical overload and oxidative stress that are induced by exercise can result in reduced concentrations of these proteins [9]. Moreover, Hsp70 is produced by cells in response to several pathological and physiological stressors, such as acute exercise, and it maintains cellular homeostasis by preventing apoptosis, influences energy metabolism, facilitates cellular processes in terms of muscular adaptation, and interacts with other signaling pathways [10]. It is worth noting that an imbalance between the anti- and proinflammatory responses might modify the iron metabolism by affecting the hepcidin expression [11]. Hepcidin is a hormone that blocks iron absorption by the intestines, and its liberation from the liver and other tissues [12]. Moreover, a high serum iron can also stimulate hepcidin expression in a mechanism that involves transferrin receptor 2 [13]. Proinflammatory cytokines stimulate hepcidin biosynthesis, which was confirmed by in vivo and in vitro studies [12,14]. Collectively, the available data indicate that the serum hepcidin levels are controlled by proinflammatory cytokines and serum iron. Hence, physical exercise may affect hepcidin biosynthesis not only by increasing the proinflammatory cytokine levels, but also by increasing the demand for iron that is required for the biosynthesis of iron-containing proteins. The above changes can also be considered to be adaptive responses to exercise. Therefore, it is to be expected that appropriately designed athletic training will reduce the blood levels of iron and hepcidin.

The maintenance of the appropriative iron status is of particular importance in young athletes. The training load in professional tennis consists of a large number of matches that are played throughout the season. The best male junior players play 21 tournaments and 56 matches in the competitive season, and female junior players play 18 tournaments and 48 matches per season [15]. Consequently, the long-lasting chronic competition period makes it difficult to plan the training load.

The game of tennis is dominated by high-intensity anaerobic exercise, with frequent changes of running direction that are interspersed with rest periods or activities of low intensity, and with an average match duration of 2–3 h [4,16]. High-intensity exercise and the resulting fatigue may induce emotional distress in young players because of their lack of experience in playing at the competitive level and high expectations from both themselves and their parents. Children, and even young teenagers, have lower anaerobic capacities than adults because of relatively low lactate dehydrogenase activity, which results in a low capacity to produce lactate [17]. The adaptation to the physical training mostly involves an increase in the synthesis and the activity of enzymes, both those that contain iron and those that do not [18]. Therefore, monitoring all of the factors that support induced physiological as well biochemical adaptive changes might help the players avoid overreaching or overtraining [19]. Different variables have been suggested as relevant for diagnosing overtraining [3]. However, research on confirmative variables is scarce. Unfortunately, there is a lack of information and comprehensive analyses on the impact of the training load on tennis players aged 12–14 years. Most of the published papers concern boys aged 14–15 years and older who play at the national level. Currently, no extensive studies are available concerning young athletes that play at the international level. Only limited studies, and only those concerning the anthropometric characteristics of the best players (girls and boys) who took part in the Davis Junior Cup and the Fed Junior Cup, have been published [20].

Consequently, the aim of the current study was to examine the effect of a training load that was imposed by two different training camps (i.e., the aim was to focus on physical conditioning or on the improvement in specific technical tennis skills, and on the immunological responses, including the iron metabolism, of young tennis players). Because of the fact that a single measurement of a biomarker does not allow for a precise determination of an individual’s health status [21], the group of biomarkers that are associated with iron metabolism, and the pro- and anti-inflammatory proteins that were induced by specific physical workloads, were assessed.

## 2. Materials and Methods

### 2.1. Overview

This follow-up study was conducted during two sports camps (the preparatory training period). The first camp (8 days) focused on physical conditioning; the second camp (14 days) focused on the improvement of technical skills. The second camp took place one month after the first camp. Blood samples were obtained and analyzed at the beginning and at the end of each camp to evaluate the cumulative effects of the training camps on the biochemical indices. All the participants were housed in the same accommodation and followed the same training schedule and a balanced diet. The daily energy content of the food did not exceed 3800 kcal. The recommended protein dose varied from 1.1 to 1.3 g/kg of body mass. No iron supplements were used during the camps, whereas the dietary iron supply was controlled and was similar during the two camps (the diet was set by the same dietitian). The training load intensity was determined on the basis of the heart rate. Details of the training loads, the exercise types, and the rest periods are presented in Table 1 and Table 2. All of the subjects had the same type of rest, which included swimming, walking, or wellness (sauna/short exposure and gentle massage). Coaches supervised the recovery process so that it had a very low intensity.

The two applied training camps that were incorporated into the study had different goals. The training load during the first camp focused on physical conditioning, and the intensity ranges were based on the individual’s heart rate. The dominant intensity of each training session ranged from low (8.5% of total time) to moderate (53% of total time).

The training load during the second camp was focused mainly on the development of specific technical skills. The intensity of each training session ranged from moderate (57% of total time) to high (24% of total time). In addition, some forms of stretching and gymnastics were performed, which was followed by training before noon, and in an afternoon session (Table 1).

### 2.2. Subjects

Highly ranked (singles national rankings for Poland: 1–30) young tennis players (12–14 years old) took part in the study (male: n = 14; body mass: 51.4 ± 5.9 kg; and height: 165.05 ± 5.6 cm; and female: n = 12; body mass: 52.6 ± 6.9 kg; and height: 165.8 ± 4.5 cm). The players were under the care of the same national tennis coaches. The research procedure was approved by the Bioethical Committee (KB-26/14) and the parents gave written consent for their children to participate in the study. Furthermore, the parents received written reports on the conducted research as well as individual suggestions as to the next steps to take.

### 2.3. Blood Sampling and Cytokine Analysis

Blood samples were taken from the antecubital vein and were deposited into single-use containers that contained the anticoagulant, EDTAK_2_. Following collection, the samples were immediately placed at 4 °C. Within 20 min of sampling, the samples were centrifuged at 3000× *g* and 4 °C for 10 min. Aliquots of the plasma were stored at −80 °C. The blood was collected at rest early in the morning. The serum IL-6, IL-10, and TNF-α levels were determined by enzyme immunoassays by using commercial kits (R&D Systems, Minneapolis, MN, USA). The detection limits for TNF-α, IL-6, and IL-10 were 0.039, 0.500, and 0.038 pg⋅mL^−1^, respectively. The average intra-assay coefficient of variability (CV) is <8.0% for all the cytokines. The serum heat shock protein Hsp27 and Hsp70 levels were evaluated using ELISA kits (Calbiochem, San Diego, CA, USA, and Stressgen, La Jolla, CA, USA, respectively). The kit detection limit is 0.2 ng⋅mL^−1^, and the intra-assay CV is <5%. The exercise-induced changes in the plasma volume during the study period were calculated by using a formula that was developed by Van Beaumont and coworkers [22].

### 2.4. Monitoring of Training Intensity and Perceived Recovery Status (PRS)

The exercise intensity was monitored by heart rate measurement and the percentage of the maximal values with regard to the information that was obtained from individual coaches or parents. The intensities of each training unit are presented in Table 2. The PRS scale was used to assess the recovery statuses. The players were given standardized instructions that explained how to interpret the PRS scale. The exercise protocol was designed to expose the subjects to exercise sessions in which they were not fully recovered or able to optimally perform. Each individual was asked to perform an identical exercise session in which the recovery duration from the preceding bout was manipulated. This enabled an improved inference of the PRS utility with the “under-recovered” subjects, and then, progressively, with the more recovered subjects [23].

The detailed structure of the training sessions is presented in Table 2. Some types of training were repeated during the two camps, but some differences resulted from the different goals of the camps. Furthermore, the second camp was twice as long as the first camp and, hence, there were considerable differences in terms of the training hours. The first camp was dominated by sessions that focused on the development of coordination, agility, and accuracy, as well as conditioning exercises, and that involved team sports activities of high intensities. The second camp was mainly dedicated to training that was focused on the footwork on the tennis court, as well as the development of the coordination, agility, and accuracy of movement.

### 2.5. Statistical Analysis

The measures related to the blood parameters were analyzed by using a spreadsheet for a post-only crossover trial [24], and the effects were interpreted by using magnitude-based inference. All data were log-transformed to reduce the bias that arises from the nonuniformity of the error [25]. The means of the score changes, the standard deviations of the score changes, and the effects (differences in the changes of the means and their certainty limits) were back-transformed to percentage units. To improve the precision of the estimates, the mean changes were adjusted to the log-transformed baseline mean. The magnitudes of the effects were also evaluated using the log-transformed data by standardizing the use of the standard deviation of the baseline values. The threshold values for the assessment of the magnitudes of the standardized effects were 0.20, 0.60, 1.2, and 2.0 for small, moderate, large, and very large effects, respectively. The uncertainty for each effect was expressed as a 90% confidence limit, as well as a probability of the true effect. These probabilities were used to make a qualitative probabilistic nonclinical inference about the true effect: if the probability of the effect being a substantial increase or decrease was >5% in both cases (equivalent to 90% confidence interval (CI) overlapping thresholds for a substantial increase and decrease), the effect was reported as unclear; otherwise, it was considered clear and was assigned the relevant magnitude value, with the qualitative probability of the true effect being a substantial increase, a substantial decrease, or a trivial difference (whichever outcome had the largest probability). The following scale for interpreting the probabilities was used: 25–75%, possible; 75–95%, likely; 95–99.5%, very likely; and >99.5%, most likely [24]. This study involved the assessment of substantial changes in nine measures. To maintain an overall error rate of <5% for declaring one or more changes to have an opposite magnitude (a substantial decrease instead of an increase, and vice versa), the effects were also evaluated as clear or unclear with a threshold of 5%/5 equivalent to the consideration of the overlap of the substantial values with a 98% CI. The relationships between the changes in the blood parameters were also calculated using the Pearson correlation coefficient. The outcomes were expressed as values with a 90% CI. The typical scale for correlation coefficients (0.1, 0.3, 0.5, 0.7, and 0.9 for low, moderate, high, very high, and nearly perfect, respectively) was used [25].

## 3. Results

We hypothesized that the different workload intensities during both camps would affect the immunological responses. Therefore, we evaluated the changes in the immunological biomarker levels at two time points: at baseline and at the end of each camp. Data are presented separately for each camp and sex in Table 3. A one-month break between the camps and a return to individual training resulted in a renewed increase in the level of proinflammatory TNFα.

The applied training programs (conditioning and technical) induced increases in the IL-6 levels in the female players. The change was smaller in the male players than in the female players and was accompanied by a small decrease in the IL-10 levels. Although the physical workload during the camps was varied from low to high, it induced a substantial decrease in the proinflammatory TNF-α levels, independently of the sex. At the same time, significant changes were apparent in the levels of hepcidin and iron. They both dropped, and the changes were more pronounced in the female players than in the male players. The changes in the hepcidin levels did not significantly correlate with the blood ferritin levels. Furthermore, the Hsp27 levels were substantially elevated only at the end of the technical camp (a moderate effect). The changes in the Hsp70 levels were unclear and small in response to the workloads at both camps.

The IL-6 levels in the male players were elevated (a moderate effect, most likely) at the end of the technical camp and were small at the end of the conditioning camp. The IL-10 levels showed a similar tendency, with an unclear increase at the end of the conditioning camp (29%), and a moderate very likely change after the technical camp (67%). A moderate decrease in the TNF-α levels was observed only at the end of the conditioning camp, with a small decrease in the hepcidin levels (8%) after both camps. A large drop in the blood iron levels (30%, most likely) was only observed at the end of the first camp, with a trivial reduction in the ferritin levels. The changes in the hepcidin levels significantly correlated with the blood ferritin levels after the conditioning camp (correlation coefficient of 0.6). A small change in the Hsp27 levels was noted only at the end of the technical camp.

In all the athletes, the TNF-α levels dropped at the end of the conditioning camp (30%, a moderate effect, likely) and at the end of the technical camp (16.7%, a small effect, likely) with a similarly small and likely decrease in the hepcidin levels (approximately 8%) after both camps. The IL-6 levels showed an opposite tendency, with a moderate very likely increase at the end of the conditioning camp (46%), and a moderate/large most likely change after the technical camp (89%). A very large drop in the blood iron levels was observed at the end of the conditioning camp (44%, most likely), with a trivial reduction in the ferritin levels (5%). A decrease in the blood iron levels after the technical camp was moderate, very likely (19%), with a possible but small drop in the ferritin levels (12%). A small/moderate elevation in the Hsp27 levels was observed only at the end of the technical camp (36%). The changes in the hepcidin levels did not significantly correlate with the blood ferritin levels. All data are presented in Table 4. Furthermore, the fatigue scale analysis revealed that the fatigue assessments at the beginning and following each camp were the same. The average values (± SD) for the boys were initially 6 ± 1, 5 ± 1 after the conditioning camp, and 5 ± 1 after the technical camp. For the girls, the fatigue assessments at the same time points were 6 ± 1, 5 ± 1, and 6 ± 1, accordingly.

## 4. Discussion

The data presented in the current study clearly indicate that the evaluated training camps reduced the proinflammatory responses and had an anti-inflammatory effect on the participants. These changes were accompanied by decreased serum iron and hepcidin levels. According to several studies, a decrease in the iron stores in athletes is provoked by inflammation that results from overreaching [26]. In the current study, the physical workload induced the opposite response, with decreased inflammation marker levels, and these changes were associated with the drop in the serum iron levels [27]. Typically, high iron levels are deemed to be beneficial, and this opinion persists among parents and coaches. The other face of the proinflammatory effect of iron is less recognized. Iron stimulates the synthesis of proinflammatory cytokines and induces oxidative stress [27]. In cells, iron is stored in a safe way by ferritin, which protects iron from free radical reactions and is not a stimulus for an inflammation process. However, it has been shown that, during stress conditions, ferritin can be degraded in lysosomes and proteasomes [28,29], which can lead to the liberation of iron, which, in turn, can induce the pro-inflammatory response [30]. Thus, even if the stored iron is in the normal range, it can still be toxic during stress conditions. The data presented herein reveal that the drop in the serum iron levels did not result from augmented inflammation but, conversely, that it was associated with reduced inflammation that was expressed in the decline in the TNFα.

Adaptation to physical training is associated with enhanced erythropoiesis and the biosynthesis of iron-containing proteins in the skeletal muscle [31]. Hence, one may speculate that the observed reduction in serum iron levels is an outcome of an adaptive response. The elevated TNFα levels before the training camps might suggest that the balance between exercise and rest was not sufficient (to reduce the inflammatory response) in the in-home conditions. This is particularly important in tennis, where a personal approach to training is dominant from an early career stage. Because of the predominantly individual nature of the competition, young tennis players, together with their parents and coaches, may be inclined to focus the training regime on this discipline alone. Of note, the parents’ expectations for the players’ success and for their advancement in the ranking entices them to train their children individually. Interestingly, the player assessments of fatigue, which are based on the scale that was proposed by Laurent et al. [23], were similar at the beginning and at the end of the camps, which might indicate that the recovery at home was not sufficient. However, the data presented herein suggest that a structured group approach to training may be valuable, particularly at an early career stage. As a result of attending both training camps, the TNFα levels decreased significantly, which confirms the benefits of a controlled workload, diet, and sleep program. Thus, reversed results of the proinflammatory marker within a short period of time might indicate that it is a prevention against the occurrence of overreaching.

Currently, the data regarding elite children or adolescent tennis players are limited. To date, studies on the subject have mainly focused on anthropological or physical performance assessments, which analyze the differences in the body height and the humerus and femur widths [20], or on the physiological demands of a tennis match, which are illustrated by the blood lactate levels and the heart rate [32]. Mendez-Villanueva et al. [33] investigated the relationship between metabolic factors (i.e., blood lactate levels) and perceptual factors (ratings of perceived exertion) and they report that the intensity of a tennis match is higher than expected. A tournament match requires high efficiency of both aerobic and anaerobic metabolism. As has been well documented, the glycolytic energy system is not highly productive during puberty [34]. Personal training, which is dominant in tennis, makes it impossible, or limits physical activity in the form of team games, although such activities are a good way of improving the anaerobic energy metabolism, coordination, or agility. Although the glycolytic enzyme capacity was not measured in the current study, both training camps involved high-intensity activities to enhance anaerobic metabolism. Furthermore, both training programs resulted in the proinflammatory response, which is a known contributor to an improved adaptation to exercise [1,35].

The training program that was followed during the camps reduced the TNFα levels and, at the same time, induced an IL-6 level increase in the tennis players. According to one study, IL-6 stimulates hepcidin biosynthesis [36]. Surprisingly, the increases in the IL-6 levels that were observed herein were not accompanied by elevated serum hepcidin levels. This indicates that the decrease in the serum iron levels was not caused by attenuated absorption but, rather, by its possible metabolic utilization.

Diminished iron levels have been noted during the tournament season in older male tennis players [29], with changes in the IL-6 levels accompanying an increase in the levels of the anti-inflammatory cytokine, IL-10 [37]. IL-6 is described as an “energy allocator” in response to metabolic stress in several tissues [38]. In the current study, a similar immune response was apparent mainly in the male players. However, changes in the hepcidin and iron levels were more pronounced in the female players than in the male players. This suggests that, in the former, the levels of the proteins that are responsible for iron metabolism are much more sensitive markers of physical workloads than in the latter. Overall, the observed changes confirm that the elevated physical workload increased the iron demand.

Of note, the Hsp27 levels significantly increased in response to the technical camp training only in the female players. It has been proposed that blood Hsp27 plays a direct role in protecting against the oxidative stress that is induced by exercise and hypoxia [39]. The physical workload that was imposed during the technical camp included more high-intensity exercise than that in the conditioning camp. According to a previously published report, 3 d of controlled physical workload elicited an increase in the Hsp27 levels [7]. The more pronounced responses to the technical camp in the female group may be due to the fact that girls in adolescence generally perform better than males during balance tasks. Among boys, a transient period of motor incoordination very often occurs during the adolescent growth spurt, which disturbs performance tasks that require balance [40].

The low Hsp27 levels that were observed herein at baseline could be associated with the forced physical workload before the training camps and possible overreaching, as is indeed confirmed by the fatigue scale assessment. The observed changes in the Hsp27 levels could have contributed to the improvement in the wellbeing and ameliorate recoveries of the players. It is puzzling that the camp participants rated their fatigue before and after their camp participation as equal. Mekari et al. [41] report that selectively executive functions are sensitive to high-intensity interval training. It cannot be ruled out that the increase in the Hsp levels was caused not only by appropriately planned workload and rest but also by the effort type. Indeed, Periard et al. [42] report that the immunoinflammatory release of extracellular Hsp27 in response to exercise might be exercise-duration- and intensity-dependent.

The current study has some limitations. First, no control group was devised. That is mainly because children playing recreationally are not able to achieve the same intensity as the group that was studied by us. Second, no special fitness tests were performed. During the training camps (national team groupings), the coaches do not want to spend the days on testing but prefer to focus on improving the skills of tennis players. Nonetheless, the limitations do not affect the results and their interpretation. In summary, the presented study reveals that both reduced proinflammatory cytokines and serum iron levels could be used as the markers of a properly designed physical workload. Professional sport often prompts a shift towards an individual approach to training, and already at the early, prepubescent, and adolescent ages. This is particularly true for tennis players.

## 5. Conclusions

The present study reveals the importance of monitoring the iron status and the low-grade systemic inflammation simultaneously. It is crucial to note that impairment in iron metabolism, very often, is not related to the amount of iron consumption but to other factors. This study demonstrates that properly designed training and rest can reverse, within a short period of time, some inflammatory outcomes, and that it can have a beneficial effect on iron metabolism.

## Figures and Tables

**Table 1 ijerph-19-03486-t001:** Training programs during first and second camps.

Days	Before Lunch	Training Intensity	After Lunch	Training Intensity
**First camp (conditioning capacity improvement)**				
1	Training A (7:45 a.m.–8:15 a.m.) Training D (9:30 a.m.–12:00 p.m.)	Low Moderate	Training I (3:00–5:00 p.m.)	Moderate
2	Training E (9:30–12:00 p.m.) Training F (12:00–12:30 p.m.)	High Moderate	Training A (3:15–4:15 p.m.) Training H (4:15–5:00 p.m.) Training C (5:00–6:00 p.m.)	Low Moderate High
3	Training A (7:45–8:15 a.m.) Training B (9:00–11:00 a.m.) Training D (11:30 a.m.–12:00 p.m.)	Low Moderate Moderate	Training I (3:00–4:30 p.m.) Training E (5:00–6:00 p.m.)	Moderate Low
4	Training A (7:45–8:15 a.m.) Training E (9:30–10:30 a.m.) Training G (11:00 a.m.–12:30 p.m.) Training D (11:45 a.m.–12:30 p.m.)	Low Low Moderate Moderate	Training E (3:00–4:00 p.m.) Training J (4:00–5:00 p.m.)	High Low
5	Training A (7:45–8:15 a.m.) Training B (9:30–11:30 a.m.) Training D (12:00 p.m.–12:45 p.m.)	Low Moderate Low	Training H (3:00–4:00 p.m.) Training C (4:30–5:30 p.m.)	Moderate Moderate
6	Training A (7:45–8:15 a.m.) Training B (9:30–11:30 a.m.) Training A (11:45 a.m.–12:45 p.m.)	Low Moderate Low	Training I (3:00–4:30 p.m.) Training D (5:00–5:45 p.m.)	Moderate Moderate
7	Training A (7:45–8:15 a.m.) Training B (9:30–11:30 a.m.) Training C (11:45 a.m.–1:00 p.m.)	Low Moderate High	Training H (3:00–4:30 p.m.) Training J (6:00–6:45 p.m.)	Low Low
8	Training A (7:45–8:15 a.m.) Training C (9:30–11:00 a.m.)	Low High	-	-
	Total (%)	Low—28.6	Moderate—53.4	High—18.0
**Second camp (technical abilities improvement)**				
1	-	-	Training C (8:00–9:00 p.m.)	High
2	Training A (7:55–8:15 a.m.) Training I (9:30–10:50 a.m.)	Low Moderate	Training E (3:00–5:00 p.m.)	High
3	Training F (9:30–11:00 a.m.)	Moderate	Training A (3:30–4:00 p.m.) Training C (4:00–5:00 p.m.)	Low High
4	Training A (7:55–8:15 a.m.) Training F (9:30–11:00 a.m.)	Low Moderate	Training A (3:30–4:00 p.m.) Training I (4:00–5:20 p.m.) Training E (8:00–9:00 p.m.)	Low Moderate High
5	Training A (7:55–8:15 a.m.) Training D (9:30–11:00 a.m.) Training C (11:30 a.m.–12:20 p.m.)	Low Moderate High	Training H (3:00–4:00 p.m.) Training K (4:30–5:30 p.m.) Training E (6:00–7:00 p.m.)	Moderate Moderate High
6	Training A (7:55–8:15 a.m.) Training F (9:30–11:00 a.m.) Training D (11:45 a.m.–12:30 p.m.)	Low Moderate Moderate	Training E (3:00–4:00 p.m.) Training J (4:00–5.00 p.m.)	High Low
7	Training A (7:55–8:15 a.m.) Training C (11:30 a.m.–12:20 p.m.)	Low High	Training I (3:00–4:20 p.m.) Training A (5:30–6:30 p.m.)	Moderate Low
8	Training A (7:55–8:15 a.m.) Training F (9:30–11:00 a.m.) Training D (11:45 a.m.–12:30 p.m.)	Low Moderate Moderate	Training H (3:00–4:00 p.m.) Training K (4:30–6:00 p.m.)	Moderate Moderate
9	Training A (7:55–8:15 a.m.) Training C (11:30 a.m.–12:20 p.m.)	Low High	Training I (3:00–4:20 p.m.)	Moderate
10	Training A (7:55–8:15 a.m.) Training J (10:00–12:00 a.m.)	Low Low	-	-
11	Training A (7:55–8:15 a.m.) Training F (9:30–11:00 a.m.) Training C (11:30 a.m.–12:30 p.m.)	Low Moderate High	Training H (3:00–4:00 p.m.) Training A (6:00–7:00 p.m.)	Moderate Low
12	Training A (7:55–8:15 a.m.) Training F (9:30–11:00 a.m.) Training D (11:45 a.m.–12:30 p.m.)	Low Moderate Moderate	Training I (3:00–4:20 p.m.) Training K (5:30–6:30 p.m.)	Moderate Moderate
13	Training A (7:55–8:15 a.m.) Training C (11:30 a.m.–12:30 p.m.)	Low High	Training H (3:00–4:00 p.m.) Training J (6:00–6:45 p.m.)	Moderate Low
14	Training C (9:00–10:00 a.m.)	High	-	-
	Total [%]	Low—19.7	Moderate—56.6	High—23.7

**Table 2 ijerph-19-03486-t002:** Comparison of times for each type of training during two camps.

Training	Time (min)	
	Conditioning Camp	Technical Camp
Training A: Stretching exercise; hold–relax technique; gymnastics; very low intensity.	330	400
Training B: Agility games with tennis balls (main emphasis on coordination, agility, accuracy); average heart rate at 60–70% max.	480	-
Training C: Conditioning exercise, team sports; short games with short periods with high intensity; average heart rate at 80–90% max. Most vital elements during games were appropriate mechanical performance of exercises and scoring maximum number of points; heart rate corresponding to the respective trials, at 60–100% of the max.	285	450
Training D: Athletics; endurance; continuous distance running; average heart rate at 70% to 80% of max.	270	225
Training E: International Physical Fitness Test–speed; jumping; endurance; power of the hand; strength of hands; agility; strength of the abdominal muscles; flexibility.	330	300
Training F: Foot work; technique skills on the tennis court; average heart rate at 80–100% max.	30	540
Training G: Throws; jumping—techniques of execution; average heart rate at 60–70% max.	90	-
Training H: Stabilizing training.	195	240
Training I: Functional exercises with bands; average heart rate at 50–60% max.	300	400
Training J: Wellness; swimming; recovery; low intensity.	105	225
Training K: Motor coordination; average heart rate at 80–90% max.	-	210
Total (min)	2415	2990

**Table 3 ijerph-19-03486-t003:** Immunological responses induced by physical workload after both camps in female (n = 12) and male (n = 14) youth athletes.

	Females			Males		
		After Camp Change (%)			After Camp Change (%)	
	Baseline Mean ± SD	Mean Change; ±90%CL	^a^ Effect	Baseline Mean ± SD	Mean Change; ±90%CL	^a^ Effect
**Conditioning camp**						
TNFα (ng·mL^−1^)	1.42 ± 0.68	−41% ±37%	moderate ↓*	1.53 ± 0.48	−19% ±14.6%	moderate ↓**
IL-6 (pg·mL^−1^)	0.77 ± 0.26	66%; ±54%	**large** ↑*******	1.15 ± 1.01	31.4%; ±30.2%	small ↑**
IL-10 (pg·mL^−1^)	1.11 ± 1.38	−39%; ±32%	**small/moderate** ↓******	0.56 ± 0.20	28.7%; ±16.4%	unclear
HSP 70 [ng·ml^−1^]	0.09 ± 0.07	−8.0%; ±33%	unclear	0.15 ± 0.35	48%; ±73%	small ↑*
HSP 27 [ng·ml^−1^]	11.3 ± 6.0	7.9%; ±31%	unclear	8.9 ± 4.8	−3.4%; ±15.3%	unclear
Hepcidin [ng·mL^−1^]	8.8 ± 1.8	−6.9%; ±3.2%	**small** ↓******	7.9 ± 2.2	−8.1%; ±4.2%	small ↓**
Iron [μg·dL^−1^]	105 ± 18	−52%; ±6.7%	**very large** ↓********	116 ± 25	−37%; ±10.8%	**large** ↓********
Ferritin [μg·dL^−1^]	32 ± 14.3	−3.5%; ±24%	unclear	37 ± 22	−6.9%; ±17.0%	trivial
**Technical camp**						
TNFα (ng·mL^−1^)	1.49 ± 0.62	−38% ±15.4%	**moderate** ↓******	1.59 ± 0.56	−6.8% ±19.8%	trivial
IL-6 (pg·mL^−1^)	0.83 ± 0.31	88%; ±30%	**large** ↑********	1.12 ± 0.96	90%; ±31.7%	**moderate** ↑********
IL-10 (pg·mL^−1^)	1.05 ± 1.21	−18.8%; ±25%	small ↓*	0.63 ± 0.17	67%; ±52%	**moderate** ↑*******
HSP 70 [ng·ml^−1^]	0.10 ± 0.06	43%; ±55%	small ↑*	0.21 ± 0.29	46%; ±89%	small ↑*
HSP 27 [ng·ml^−1^]	10.9 ± 5.0	61%; ±36%	**moderate** ↑*******	7.9 ± 6.1	17.0%; ±22%	small ↑*
Hepcidin [ng·mL^−1^]	8.6 ± 1.9	−7.7%; ±3.2%	**small** ↓******	7.7 ± 2.3	−7.4%; ±10.5%	small ↓*
Iron [μg·dL^−1^]	99 ± 24	−30%; ±6.7%	**large** ↓********	112 ± 31	−8.2%; ±16.2%	unclear
Ferritin [μg·dL^−1^]	30 ± 11.9	−16.4%; ±21%	small ↓*	33 ± 17	−7.9%; ±15.9%	trivial

90%CL: 90% confidence limit; ↑: increase; ↓: decrease. ^a^ magnitude thresholds: <0.20, trivial; 0.20–0.59, small; 0.60–1.19, moderate; 1.2–1.99, large; 2.0–3.99, very large. Likelihood that the true effect is substantial: * possible; ** likely; *** very likely; **** most likely. Effects in bold are also clear at 0.5% level.

**Table 4 ijerph-19-03486-t004:** The immunological responses induced by physical workload after conditioning and technical camp in all youth athletes (females and males; n = 26).

	Conditioning Camp			Technical Camp		
		After Camp Change (%)			After Camp Change (%)	
	Baseline Mean ± SD	Mean Change; ±90%CL	^a^ Effect	Baseline Mean ± SD	Mean Change; ±90%CL	^a^ Effect
TNFα (ng·mL^−1^)	1.48 ± 0.57	−30% ±18.9%	**moderate** ↓******	1.54 ± 0.59	−16.7% ±14.5%	**small** ↓******
IL-6 (pg·mL^−1^)	0.97 ± 0.77	46%; ±26%	**moderate** ↑*******	1.00 ± 0.64	89%; ±46%	**moderate/large** ↑********
IL-10 (pg·mL^−1^)	0.82 ± 0.96	−8.8%; 24%	unclear	0.85 ± 0.69	19.8%; ±25%	small ↑*
HSP 70 [ng·mL^−1^]	0.12 ± 0.26	18.6%; ±35%	trivial	0.16 ± 0.18	44%; ±48%	small ↑*
HSP 27 [ng·mL^−1^]	10.0 ± 5.4	1.6%; ±15.1%	unclear	9.4 ± 5.4	36%; ±21%	**small/moderate** ↑*******
Hepcidin [ng·mL^−1^]	8.3 ± 2.0	−7.5%; ±2.8%	**small** ↓******	8.2 ± 2.0	−7.6%; ±5.4%	**small** ↓******
Iron [μg·dL^−1^]	111 ± 23	−44%; ±6.9%	**very large** ↓********	106 ± 28	−19.2%; ±8.8%	**moderate** ↓*******
Ferritin [μg·dL^−1^]	34 ± 18.7	−5.4%; ±13.0%	trivial	32 ± 14.5	−12.0%; ±12.0%	**small** ↓*****

90%CL: 90% confidence limit; ↑: increase; ↓: decrease. ^a^ magnitude thresholds: <0.20, trivial; 0.20–0.59, small; 0.60–1.19, moderate; 1.2–1.99, large; >2.0, very large. Likelihood that the true effect is substantial: * possible; ** likely; *** very likely; **** most likely. Effects in bold are also clear at 0.5% level.

## Data Availability

The data presented in this study are available on request from the corresponding author. Moreover, data were sent to Ministry of High Education as final report.

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
