# Peer review of "Planned Physical Workload in Young Tennis Players Induces Changes in Iron Indicator Levels but Does Not Cause Overreaching"

_ijerph, 2022, doi:10.3390/ijerph19063486_

Round 1

Reviewer 1 Report

The paper “Planned physical workload in young tennis players induces changes in iron indicator levels but does not cause overreaching” is proposed to fill the lack of information and comprehensive analyses on the impact of training load on tennis players aged 12–14 years. The authors focus their attention on several inflammatory and pro-inflammatory indicator as, TNF-alfa, IL-6, IL-10 and serum markers (as Hsp 27, Hsp 70, Iron, Ferritin and hepcidin), that should be affected during the physical workload under evaluation. The authors shown interesting variations in the level of pro-inflammatory and anti-inflammatory markers, together with a reduction of the level of iron in the samples harvested after both conditioning and technical camp.

However, the work presents several concerns that will be listed below.

MAJOR CONCERNS

  1. The central message presented on this work is that through the use of planned physical workload, in young tennis players, it is possible to affect the inflammation process and the iron level, so to improve not only the agonistic training method, but also to have a tool for the evaluation of the post-training recovery. These assumptions are based on the analysis of a limited cohort, and maybe more importantly on a strong lack of control group (as underlined by the authors themself). It is understandable the difficult to enlarge the cohort of highly ranked young tennis players; nevertheless it would have been interesting to analyze, in an untrained group, if the effect observed is connected to the workload itself or maybe it was correlated with a general change of environment, habits and diet compared to the domestic one. Moreover, it would have been useful to have a control group of 12-14 years subjects analyzed for the same markers following the same timing, so to have an evaluation of the variability of such marker in a unperturbed-like cohort;

  2. In the last few years several papers expressed doubts about the use of magnitude-based inference (MBI), in particular mostly used in sport science and medicine, see Lohse KR et al. (2020) Systematic review of the use of “magnitude-based inference” in sports science and medicine. PLoS ONE 15(6): e0235318. MBI developer also made changes to the statistical method after the criticisms received, releasing a “Magnitude-based decisions” (MBD) method (Hopkins WG. (2019). Magnitude-based decisions. Sportscience 23, i-iii) without, however, dispelling doubts. I understand that several studies are analyzed using MBI, however, seen the strong effect presented by the authors, in particular on IL-6 and iron levels, it would be preferable to evaluate the robustness of such data with a different statistical analysis;

  3. In M&M was explained that the relationships between changes in blood parameters were also calculated using the Pearson correlation coefficient, however, with the exception of the reference to the correlation of hepcidin levels with blood ferritin after the conditioning camp, none of this data are shown;

  4. As described by the authors the levels of some of the markers took in exam presented a very “peculiar” baseline before each camp. Are these “anomalous” baseline levels already described in literature, even if in older tennis players? While the levels before the conditioning camp are explained by the authors with an intensively activity, in my opinion it is important to asses which event may have restored the level at about the same baseline before the beginning of the technical camp. It is again correlated with the exacerbation of individual training or should be explained with the return to a normal environment and different habits and diet? Please address this questions.

MINOR CONCERNS

  1. The sentence in line 28 does not make sense;

  2. Abbreviation of proteins should be formatted within the same font and style in all the text, including table and figures;

  3. “The whole family of HSP proteins protect against aggregation of aberrantly folded proteins and promote their return to their native confirmations.“ in line 56 should have a reference;

  4. The sentence in line 57-59 is poorly explained, needs to be changed;

  5. In line 69 “Proinflammatory cytokines stimulate hepcidin biosynthesis in vitro [12]” there is a wrong reference or it is absolutely wrong to use in vitro in the sentence;

  6. The sentence between line 81-92 is too narrative, not useful to the informativity of the text;

  7. In line 145 it is explained the quantification assays of serum irisin, that is shown in no following part of the text and tables, nevertheless there is no elucidation of the quantification methods for Hepcidin, Iron and Ferritin, that is so central for the paper, should be added;

  8. Table 3 is doubled;

  9. In line 237-239 both percentage are referred to “the conditioning camp”, there is an error;

  10. The sentence in line 269 does not make sense;

  11. The sentence in line 319-321 should be deleted, it is not informative and distractive for the discussion;

  12. There is a typo in line 347;

  13. Conclusion should be more articulated.

Author Response

Dear Reviewer

We would like to thank for your work on our manuscript and the valuable comments on it. We have attempted to satisfy the requests in order to improve the quality of the paper. Sincerely Ewa Ziemann

Reviewer 2 Report

I want to thank to the authors their effort to shed some light on an interesting –and understudied– topic in a very unconsidered group of population. They presented a nice work carefully elaborated. With that said, I see several concerns that need the attention of the authors to be addressed, some of them of major importance, I hope the following remarks can help the authors to improve their manuscript.

My main concern is about the concept of overreaching and overtraining. I think that the concept should be well defined, controlled and used in this work https://www.thieme-connect.com/products/ejournals/abstract/10.1055/s-2007-989264.

Other concerns:

  • Line 36: the first sentence can be misleading. Excessive exercise by definition leads to unwanted outcomes. This can happen in both, technical and physical domains.
  • Line 40: I do not really think that the purpose would be avoid changes in hematological indicators, but to prevent overreaching.
  • Line 87: Dr. Baiget in Spain has been doing a great job in defining cardiorespiratory demands in tennis, you can have a look to his work at: https://journals.plos.org/plosone/article?id=10.1371/journal.pone.0131304 . I would suggest to pay special attention to the Fig.2. Tennis is an aerobic exercise with some moments between VT1 and VT2 and peaks above VT2, such others as basketball.
  • Line 126: what the wellness activity includes?
  • Lines 145-148: CV above 5% could seem high specially when using some statistical methods to detect differences.
  • Line 163: MBI can be a complementary tool to analyze data, and although I consider myself a follower of Dr. William Hopkins work, frequentist approaches, even simple as paired t-tests should be included. MBI has been found to inflate type I error rate, and when reduces type II error rate, cannot be used in isolation. Once that said, when more than one variable is measured, and analyzed in isolation, a statistical significance level correction (i.e. Bonferroni) should be used.
  • Line 193: heart rate using has to be explicated in the methods section. Heart-rate can be a good tool to infer acute physiological effects on the athlete in cardiovascular dominant activities, but not in others such short efforts of high power-muscular involvement. From this point, the difference of intensity (being the skill camp more intense, can be seen as a source of bias).
  • Table 2: both loads are different, and since the technical camp lasts longer than the conditioning camp –which should be standardized to compare­–, the effect of both can differ or be the same because loads differ. Comment.
  • Line 315: at the same age, female athletes can –and probably do– have different levels of maturity than their male counterparts. This can make results more similar to the adult players. Comment.
  • Lines 324-6: “The physical workload …” this sentence has non-sense to this reader. Is really a conditioning camp less intense than a technical-skills camp? If so, that so-called conditioning should be called in any other way.
  • Lines 349-351, although I liked the honesty of the authors declaring the limitations, there are some wrong conclusions here. The present research does not evaluate the suitableness of the program, but the physiological effect of it. At a pre- and adolescent age, tennis is can be intense, but is not professional.

Author Response

(The authors gave the same response as above.)

Reviewer 3 Report

Overall summary: This manuscript investigated the effect of two training camps on blood markers and recovery in adolescent tennis athletes. The overall findings were that the training camps (separated by one month) invoked changes in certain blood markers, where others were not impacted. The time and effort that goes into performing a study in this manner need to be commended. There are a few formatting errors that must be addressed as well as potential considerations that should be addressed before being considered prepared for publishing based on this reviewer's evaluation.

Introduction: The introduction provides sufficient background material on the topic and pertinent previous literature. Please see below for specific suggestions.

Lines 36-37 - These two sentences are referring to professional athletes specifically but this statement can be generalized to all people, and athletes in general. "Excessive" exercise based on the definition does not permit for adaptive changes or increased physical performance, it hinders it but progressive exercise is likely a more appropriate term for this concept. Consider rewording to better reflect the ultimate purpose of monitoring training loads and managing fatigue in general.

Line 61- "this" to "these", HSPO70 to Hsp70

Line 68 - Move sentence starting on line 68 discussing hepcidin's function before sentence in line 67 to allow reader to gain an understanding of this specific hormone if not known

Line 71 - as "s" to indicate

Line 108 - Due to "the" fact that a single measurement....

Methods: Overall well explained and easy to follow. Very detailed information regarding the training sessions and outline of each camp is appreciated. It may be better served to switch the order of Table 1 and Table 2 so that readers can identify what each training session designation includes before reading the training program, however not a major concern, just something for authors to consider. There needs to be a greater description of the purpose/content of the separate training camps within the methods section instead of the first paragraph of the results section. This is all part of the methods section, not the results as this is describing the design of the study. The results section is to discuss the impact of said training. Just as a supplement study (for example) would provide dosages and intervention in the methods, not in the results section.

Line 116 - replace 1 with one

Line 118 - remove "," after camp

Line 189 - need statistical reference for correlation coefficient interpretations. Although "typical" these values came from an original source that should be credited appropriately.

Table 2 - Training B - change stress to emphasis

Results: Needs to be improved. Tables are labeled poorly as well as repeated.

Include pre-post values rather than just mean change from baseline in raw form as well as based on percent change and effect size. Helps describe a more clear picture.

Remove the duplicate of Table 3.

If possible, recommend combining male, female, and total group on one table. Maybe separate the conditioning camp and technical camp to do this effectively based on spacing and reading ability.

Report actual effect sizes either within table (recommended) or within text.

Line 235 - report using r value

Discussion: Overall needs to be revised for grammar. Findings are supported by results but need to be revised to focus on the main outcomes as there is too much speculation currently. It is interesting to note that the technical camp was far more demanding compared to the conditioning camp. Is there any speculation as to why this was the case since the physical conditioning camp may not have achieved the physical conditioning necessary for development of physical qualities as it compares to technical demands.

Line 265 - 267 - Revise sentence to read more clearly. It is hard to follow and not sure of the point the authors are attempting to make which does not allow me to provide a specific suggestion for this specific revision. 

Lines 264-272 - It may be that the actual effect of iron on proinflammatory responses is not a significant finding when observed holistically which is why it may be dismissed or not often considered. Based on the evidence in this study, the levels did not approach the levels necessary for iron to have proinflammatory effects. Consider revising this statement or removing.

Line 274-remove "an"

Line 279-remove ","

Lines 284-286-Was there any reporting of training done in between camps or simply speculation/not considered?

Line 297-remove good, replace with sufficient or synonym. Good is too broad of a term.

Line 327 - change to elicited

Lines 322-331 - It is noted in the limitations that there were not any specific physical assessments completed due to coaches preference which is a part of the process and cannot be directly held against researchers. However, the speculation of balance being greater in females being the reason for the differences in males and females may be overlooking a much larger factor being developmental age of participants and potential role of males and females and the stage of puberty which was partly addressed. Balance tasks are one component and likely balance tasks played a small part of the training load experienced (<10% if motor coordination is the component in reference at moderate intensity), would that be a main influence on this observation?

Line 347 - does not affect "the results and their interpretation" I propose that in fact they may alter the interpretation if that information was known because we would have a better understanding of physical qualities of individuals that are included in each group.

Author Response

(The authors gave the same response as above.)

Reviewer 4 Report

Dear Authors,

Thank you for the opportunity to review this manuscript. In general, the manuscript is well written and sound. The introduction to the issue of impact that the training load shows in the inflammatory indicators in young tennis players is well substantiated. Also, the aims and procedure are properly described. The results are meaningful and adequately discussed with the current literature.

However, there are some parts that need reviewing before publication.

L75: The authors claim that a properly designed training program would reduce iron hepcidin blood levels. However, would this intervention lead to changes in other markers that would eventually give unwanted side effects or even masking of positive blood level parameters?

L96: What would usually be the procedure to monitor overreaching or overtraining information in youngsters? Typical blood analysis?

L105: Why would two different training styles have an impact on immunological responses? The above text in the introduction does not give a clear reason.

L128: The authors didn’t use a control group. Please explain why you choose such a study design, taking into account that a control group would have given strong evidence of results.

L130: I think the authors wanted to state “body mass” instead of body weight. Also the authors should do well to describe height as 165.0

L131: It is very unusual that both boys and girls measured the same mean value. Is that a typo?

L165: Did the authors analyze the lack of uniformity of the errors and then used a log-transformed version or did they use the log-transformed because they anticipated such non-uniformity in advance?

L191-208: I think this part should be incorporated into the methods section, rather than in the results section as the authors are not describing any measure or outcome.

L214: Please, be consistent with the number of decimal places in Table 3.

L294: Why in some cases the authors wrote “et al” and some other cases “and colleagues”?

L394: Although the authors expressed reasons why they did not use a control group, they should do well in elaborating a bit more and even hypothesizing how a study design like this could have been done with a control group if they have the ability to modulate the training camps.

L347: I think the authors wanted to state “our” than “your”.

Author Response

(The authors gave the same response as above.)

Round 2

Reviewer 1 Report

Dear authors,

thank you for addressing many of my concerns.

Despite my remaining perplexities about the MBI analysis, on which I believe it is important to open a deeper scientific debate, and understanding the impossibility at this point to enlarge the cohort of your study, I decided to recommend the acceptance of your paper, after minor revision.

line 27: change and with end;

line 30-31: period lacks commas;

line 38-41: period not clear, needs to be modified;

line 45-48: period not clear, needs to be modified;

line 100: english style not conformed, needs to be modified;

line 306-309: english style of these two periods are not correct, needs to be modified;

line 328: missed a fullstop;

line 329-330: not clear at all, needs to be reformulated;

line 355: this period is not clear at all, should be changed with: "IL-6 was described as energy allocator in response to metabolic stress in several tissues"; 

Conclusion should be improved, in particular the sentence "It is crucial to note that impairment in iron metabolism very often is not related to amount of iron consumption but to other factors" is too generic for a conclusion chapter. However, the sentence should be change in this way "This study demonstrates that properly designed training and rest can reversed, within a short period of time, some inflammatory outcomes and have beneficial effect on iron metabolism."

Author Response

Dear Editor and Dear Reviewer

We would like to thank you once again for the interest taken in our paper and the time spent on reviewing it. We would also like to thank you for the second chance to revise it. We have corrected our manuscript according suggestion of first reviewer and we hope that manuscript in the present form, it will meet the high standards of the International Journal of Environmental Research and Public Health.

Best Regards Ewa Ziemann

Reviewer 4 Report

The paper has improved significantly after the authors considered the indications.

Author Response

Dear Reviewer 

We would like to thank you once again for the interest taken in our paper and the time spent on reviewing it. 

Best regards Ewa Ziemann